# Graph Contrastive Learning with Kernel Dependence Maximization for Social Recommendation

## ABSTRACT

Contrastive learning (CL) has recently catalyzed a productive avenue of research for recommendation. The efficacy of most CL methods for recommendation may hinge on their capacity to learn representation uniformity by mapping the data onto a hypersphere. Nonetheless, applying contrastive learning to downstream recommendation tasks remains challenging, as existing CL methods encounter difficulties in capturing the nonlinear dependence of representations in high-dimensional space and struggle to learn hierarchical social dependency among users— essential points for modeling user preferences. Moreover, the subtle distinctions between the augmented representations render CL methods sensitive to noise perturbations. Inspired by the Hilbert-Schmidt independence criterion (HSIC), we propose a graph **C**ontrastive **L**earning model with **K**ernel **D**ependence **M**aximization CL-KDM for social recommendation to address these challenges. Specifically, to explicitly learn the kernel dependence of representations and improve the robustness and generalization of recommendation, we maximize the kernel dependence of augmented representations in kernel Hilbert space by introducing HSIC into the graph contrastive learning. Additionally, to simultaneously extract the hierarchical social dependency across users while preserving underlying structures, we design a hierarchical mutual information maximization module for generating augmented user representations, which are injected into the message passing of a graph neural network to enhance recommendation. Extensive experiments are conducted on three social recommendation datasets, and the results indicate that CL-KDM outperforms various baseline recommendation methods.

## CCS CONCEPTS

• **Information systems → Social recommendation.**

## KEYWORDS

Self-Supervised Learning, Contrastive Learning, Hilbert-Schmidt independence criterion, Data Augmentation, Graph Neural Networks

**Relevance:** This work is relevant to the focus of *understanding better the impact of the Web and Web technologies*, and to the track of *Graph Algorithms and Modeling for the Web*. This work proposes a graph contrastive learning with kernel dependence maximization algorithm, aiming to address the challenge of how to learn the nonlinear dependence of representations in high-dimensional space in the field of graph representations.

## 1 INTRODUCTION

In the realm of recommender systems, various collaborative filtering techniques have been utilized to map users and items into latent space [14]. Among these methods, Graph Neural Networks (GNNs) have risen as prominent frameworks within the context of collaborative filtering, effectively capturing high-order connectivity patterns between users and items [31]. However, their effectiveness hinges on plentiful high-quality labels, struggling with sparse or noisy labeled data [12].

To mitigate label-dependency issues in GNN-based recommendation, contrastive learning (CL) has rekindled interest in recommender systems [15, 18, 28]. CL's capacity to extract features from unlabeled data offers promise in addressing data sparsity naturally [3, 11]. Recent research [12-15] has leveraged CL to improve recommendation. Depending on the way in which contrasting views are generated for collaborative filtering, recent CL models can be broadly classified into: i) *Structure-Level Augmentation* [12]: augmenting user-item graphs with structural perturbations based on topology to create contrastive views and maximizes representation consistency via a graph encoder; ii) *Representation-Level Augmentation* [3]: generating contrasts through random noise added to representations; iii) *Local-Global Augmentation* [14]: reconciling local user-item embeddings with global information through local-global contrastive learning. Despite the encouraging results achieved by CL, however, they are not sufficient to deal with the nonlinear dependence of user/item representations in high-dimensional space and the modeling of social dependency due to hierarchical structures, which are essential for user preference modeling.

Firstly, the study [3] revealed that the success of contrastive learning applied to recommendation primarily stems from the uniform representation distribution, thus implicitly acting as popularity debiasing and capturing the intrinsic characteristics of users and items. Delving deeper into the study [28], we argue the key to contrastive learning lies in that it uses $L_2$ normalization to map raw data to hyperspheres for representation differentiation and uniformity. However, the original intent of $L_2$ normalization is to map the vectors onto a unit length hypersphere for similarity metrics and to remove scale effects (as shown in Figure 1). Alarmingly, this process does not change the dimensionality of the data, but only influence the scale of the data, which may lead to weak differentiation of representations in high-dimensional space. Meanwhile, $L_2$ normalization may lose the magnitude information of the original data, leading to information loss. Consequently, existing CL models cannot explicitly handle nonlinear dependence between representations in high-dimensional space. This capability is essential for capturing intricate patterns of user-item interactions. Furthermore, the literature [2] implies that the effectiveness of the classical CL model InfoNCE may stem from its implicit estimation of the kernel-based dependence.

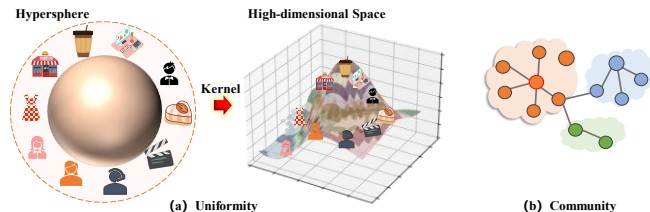

**Figure 1: Motivation examples. (a) The left diagram illustrates the uniformity of user/item representation distribution on the unit hypersphere. The right diagram enables dimension transformation by mapping the representations to high-dimensional space via kernel functions. (b) Community structures.**

Secondly, owing to the intricate user-item interactions, the influence of users' social networks on recommender systems is striking and has given rise to a plethora of social recommendation models. Hence, it is advisable to harness the higher-order social influence to enact structure-level augmentations. However, existing structure-level and local-global augmentations typically extract subgraphs to learn the connectivity patterns of a graph, potentially resulting in the loss of critical structures and the retention of noisy data. Intuitively, partitioning a graph into communities is a potent method for preserving underlying structural information [48], as it leverages the inherent community characteristics—tight internal connectivity of nodes within communities and weaker connectivity of nodes across communities—spanning local to global scale (as shown in Figure 1(b)). The community structures possess the potential to influence user preferences, underscoring the necessity of refining users' hierarchical social dependency to enhance recommendation.

To this end, an intriguing question arises: is there a means to explicitly learn the nonlinear dependence of representations in high-dimensional space and hierarchical social dependency? According to self-supervised learning from statistical dependence [2] and complex network theory, we propose to explicitly learn the kernel dependence of representations in Hilbert space and leverage community characteristics to explore the following challenges:

•**CH1**: How to learn the nonlinear dependence of representations in high-dimensional space while empowering the ability to maintain semantic consistency and variability for robust and generalizable recommendation. The Hilbert Schmidt independence criterion (HSIC) provides a solution for learning the nonlinear dependence of representations, where the sparsity of high-dimensional space makes the selection of representations critical. However, current representation-level augmentations heavily rely on designed perturbations to create contrastive views of representations. Excessive perturbations can lead to substantial deviations from the original graph semantics, while overly subtle perturbations may fail to provide augmented embeddings with adequate variability. Furthermore, if the difference between augmented representations of a model is so small to be limited to constants, the model encounters challenges in generalizing to new datasets. The model exhibits sensitivity to minor input perturbations, thereby performing poorly in coping with noisy data.

•**CH2**: How to capture the hierarchical social dependency among users by leveraging hierarchical topology while preserving community structures? While the community theory provides a

viable direction for modeling hierarchical dependency, it remains challenging to leverage hierarchical community structures and simultaneously extract social dependency from the node-level to the community-level, and to the global-level. Meanwhile, the connectivity of nodes inside and outside a community implied by the community structure may make it promising to tap into the partial order relationships between negative samples in a social network, rather than just utilizing the positive samples of the social network for recommendation.

To tackle these challenges, we propose a method of hierarchical graph contrastive learning with kernel dependence maximization (CL-KDM) which is capable of learning the nonlinear dependence in Hilbert space for robust and generalizable recommendation. Specifically, we design noise perturbations with Gumbel-distribution to generate semantically consistent and variable augmented representations, and then we learn kernel dependence maximization of the augmented representations by adding HSIC to the graph contrastive learning. To capture hierarchical social dependency, we propose a hierarchical mutual information maximization approach that maximizes mutual information from the user-level to the community-level, and from the community-level to the global-level, thereby preserving the underlying structure. Finally, we design community-guided social graph reconstruction to further enhance user embeddings.

In summary, our work makes the following contributions:
• To the best of our knowledge, we are the first to introduce HSIC into recommendation to learn the kernel dependence of augmented user/item representations in Hilbert space.
• Our theoretical analysis suggests that representation-level augmentations may be restricted to constants. Hence, we introduce HSIC as a regularizer into graph contrastive learning to enhance the robustness and generalizability of recommendation.
• We design hierarchical topology enhanced social dependency modeling and introduce the dependency to the message passing. Additionally, the community-guided social graph reconstruction is integrated to enhance user embeddings.
• Extensive experiments demonstrate that CL-KDM achieves significant improvements over state-of-the-art models. Ablation study shows that the Gaussian kernel effectively learns the kernel dependence of augmented representations in Hilbert space.

## 2 Related Work

**Social Recommendation.** Social influence theories [32] suggest that users' preferences and decisions are influenced by their friends, motivating the integration of social relationships into recommendation to overcome data sparsity. Traditional methods (e.g., Sorec [33], TrustSVD [34]) use matrix factorization to project users into latent factors, assuming users with social connections share similar interests [36]. Deep learning-based models (e.g., DiffNet [35], ESRF [30], DISGCN [36]) leverage higher-order social similarity through neural networks. Models using attention mechanisms, as represented by SAMN [37] and GraphRec [38], can dynamically assign varying levels of importance to differentiate users or items, thus capturing nuanced preferences and interactions among users. Data augmentations have been applied in recent models (MHCN [10], SMIN [39]). However, most existing social recommendation falls short of

capturing essential hierarchical social dependency and underutilizing partial order relationships of negative samples in social networks, which are vital to enhancing user preference modeling. **Graph Neural Network for Recommendation**. Given the strength of representation learning on graph-structured data, graph neural networks have been widely adopted in recommendation to model various relationships in different recommendation scenarios. For example, i) user-item interactions in collaborative filtering (e.g., LightGCN [31], NGCF [7]); ii) user connections in social recommendation (e.g., ESRF [30], DcRec [29]); iii) item-item temporal relationships (e.g., TiDA-GCN [40], MA-GNN [41]); and iv) entity-item dependency in knowledge graph-enhanced recommendation (e.g., Metakg [42]). Therefore, our CL-KDM adopts GNNs as the backbone to model the high-order collaborative relationships with the injection of hierarchical topology enhanced social dependency.

**Self-Supervised Learning**. Self-supervised learning (SSL) has garnered increasing prominence as a methodology utilized for extracting supervisory signals from data, facilitating the acquisition of representations without reliance on labels [2, 16-19]. In light of the favorable outcomes achieved by SSL in the field of graph representation learning [20], recent studies have also undertaken an exploration of its applicability in recommendation scenarios [3, 12]. These studies mainly develop self-supervision tasks from the following perspectives: i) *structure-level augmentation*: models such as SGL [12] and DCL [44] employ random node/edge dropout to create contrastive views based on user-item graph augmentation; ii) *representation-level augmentation*: These works develop representation-level augmentations by adding perturbations into inputs to regulate the uniformity of representation distribution [3, 43]; iii) *local-global augmentations*: HCCF [45] and AutoCF [14] maximize mutual information from the node-level to the global-level. Different from these works, for robustness and generalization, we learn the nonlinear dependence of augmented representations in high-dimensional space to regulate the uniformity of representations.

## 3 METHODOLOGY

This section elaborates the technical details of CL-KDM. We display the architecture of CL-KDM in Figure 2. We consider a recommendation scenario with a set of $M$ users $U = \{u_1, ... u_m, ... u_M\}$ and a set of $N$ items $V = \{v_1, ... v_n, ... v_N\}$. We define relevant inputs as below: **Definition 1**: User Social Graph $\mathcal{G}_s = \{U, E_s\}$. If two user nodes $u_m$ and $u_{m'}$ are socially connected, there is an edge $E_s$ between them in $\mathcal{G}_s$. Community Graph $\mathcal{G}_C = \{U, C, E_c\}$. $C$ denotes the set of community nodes and $E_c$ denotes affiliations between user nodes and community nodes. If $u_m$ belongs to community $c_k$, there is an edge between $u_m$ and $c_k$ in $\mathcal{G}_c$. **Definition 2:** Interaction Graph $\mathcal{G}_r$ is defined as $\mathcal{G}_r = \{U, V, E_r\}$, where $E_r$ denotes edges between $u_m$ and $v_n$. **Problem Statement**. Input: the social graph $\mathcal{G}_s$, community graph $\mathcal{G}_c$ and the user-item interaction graph $\mathcal{G}_r$. Output: scores that predict the future user-item interactions.

### 3.1 Hierarchical Topology Enhanced Social Dependency Modeling

To capture the hierarchical social dependency among users, we develop a hierarchical topology enhanced social dependency framework that maximizes the mutual information between the community graph $\mathcal{G}_c$ and social graph $\mathcal{G}_s$. We first generate node-level embedding $\boldsymbol{x}_{u_m} \in \mathbb{R}^d$ for each user in our message propagation layer. The local encoding function over $\mathcal{G}_s$ is defined as: $\mathbf{X}^* = \text{ReLU}(\widehat{\mathbf{D}}_s^{-1/2}(\mathbf{A}_s + \mathbf{I}_s)\widehat{\mathbf{D}}_s^{-1/2}\mathbf{X}\mathbf{W}_s)$, where $\mathbf{X}^* \in \mathbb{R}^{M \times d}$ is the encoded hierarchical social-aware representations for all users. $\mathbf{W}_s$ is a learnable weight matrix and $\mathbf{I}_s$ is an identity matrix. $\widehat{\mathbf{D}}_s^{-1/2}(\mathbf{A}_s + \mathbf{I}_s)\widehat{\mathbf{D}}_s^{-1/2}$ performs the message aggregation for social neighbors, and $\widehat{\mathbf{D}}_s$ represents the diagonal node degree matrix of the adjacent matrix $\widehat{\mathbf{A}}_s \in \mathbb{R}^{M \times M}$. After obtaining the node-level embeddings $\mathbf{X} \in \mathbb{R}^{M \times d}$ encoded from $\mathcal{G}_s$, we learn the community-level representation $\boldsymbol{\pi}_c \in \mathbb{R}^d$ over the $\mathcal{G}_c$ and the global-level representation $\boldsymbol{r}_s \in \mathbb{R}^d$ of $\mathcal{G}_s$. We define community (global)-level aggregation function $\mathbb{R}^{M \times d} \rightarrow \mathbb{R}^d$ with the consideration of the relationships among users, communities, and the global graph as follows:

$$\boldsymbol{\pi}_c = \sigma(\frac{\sum_{m=1}^{M} \boldsymbol{X}_m \cdot \rho_{m,m}}{\sum_{m=1}^{M} \sum_{k=1}^{K} g_{m,k}}), \boldsymbol{r}_s = \sigma(\frac{\sum_{m=1}^{M} \boldsymbol{X}_m \cdot b_{m,m}}{\sum_{m=1}^{M} \sum_{m'=1}^{M} a_{m,m'}}) \quad (1)$$

where $\sigma$ denotes the sigmoid activation function and $\rho_{m,m}$ indicates the number of communities that contain user $m$, and $g_{m,k}$ indicates the user's affiliation value with the community. $b_{m,m}$ and $a_{m,m'}$ represents the element in $\widehat{\mathbf{D}}_s$ and $\widehat{\mathbf{A}}_s$, respectively.

We bridge the node-level embedding $\boldsymbol{x}_{u_m}$ and the global-level representation $\boldsymbol{r}_s$ through community-level representation $\boldsymbol{\pi}_c$, and enhance these representations by exploring hierarchical mutual information among them. To encode the mutual dependency between community $\mathcal{G}_c$ and global graph $\mathcal{G}_s$, we train a discriminator that distinguishes between positive samples $(\boldsymbol{x}_{u_m}, \boldsymbol{\pi}_c)$, $(\boldsymbol{\pi}_{c_k}, \boldsymbol{r}_s)$ and negative samples $(\acute{\boldsymbol{x}}_{u_m}, \boldsymbol{\pi}_c), (\acute{\boldsymbol{\pi}}_{c_k}, \boldsymbol{r}_s)$ from $\mathcal{G}_c$ and $\mathcal{G}_s$ while preserving hierarchical topology. Negative instances $(\acute{\boldsymbol{x}}_{u_m}, \boldsymbol{\pi}_c), (\acute{\boldsymbol{\pi}}_{c_k}, \boldsymbol{r}_s)$ are generated based on the node shuffling strategy [27]. The positive instances $(\boldsymbol{x}_{u_m}, \boldsymbol{\pi}_c)$, $(\boldsymbol{\pi}_{c_k}, \boldsymbol{r}_s)$ and negative instances $(\acute{\boldsymbol{x}}_{u_m}, \boldsymbol{\pi}_c), (\acute{\boldsymbol{\pi}}_{c_k}, \boldsymbol{r}_s)$ are then fed into the discriminator function $\psi(\cdot): \mathbb{R}^d \times \mathbb{R}^d$:

$$\psi(\boldsymbol{x}_{u_m}, \boldsymbol{\pi}_c) = \sigma(\boldsymbol{x}_{u_m}^{\text{T}} \cdot \mathbf{W_1} \cdot \boldsymbol{\pi}_c)$$
$$\psi(\boldsymbol{\pi}_{c_k}, \boldsymbol{r}_s) = \sigma(\boldsymbol{\pi}_{c_k}^{\text{T}} \cdot \mathbf{W_2} \cdot \boldsymbol{r}_s) \quad (2)$$

where $\psi(\cdot)$ generates a probability score of node $u_m$ ($c_k$) belonging to community $\mathcal{G}_c$ (graph $\mathcal{G}_s$) given representations $(\boldsymbol{x}_{u_m}, \boldsymbol{\pi}_c)$, $(\boldsymbol{\pi}_{c_k}, \boldsymbol{r}_s)$. $\mathbf{W_1}, \mathbf{W_2} \in \mathbb{R}^{d \times d}$ are learnable matrices. We then define community-level mutual information-based loss $\mathcal{L}_{mu_c}$ and global-level mutual information-based loss $\mathcal{L}_{mu_g}$ as follows:

$$\mathcal{L}_{mu_c} = -\frac{1}{N_{pos}^u + N_{neg}^u}\Big(\sum_{m=1}^{N_{pos}^u} \rho(\boldsymbol{x}_{u_m}, \boldsymbol{\pi}_c) \cdot \log\psi(\boldsymbol{x}_{u_m}, \boldsymbol{\pi}_c)$$
$$+ \sum_{m=1}^{N_{neg}^u} \rho(\acute{\boldsymbol{x}}_{u_m}, \boldsymbol{\pi}_c) \cdot \log[1 - \psi(\widetilde{\boldsymbol{x}}_{u_m}, \boldsymbol{\pi}_c)]\Big)$$
$$\mathcal{L}_{mu_g} = -\frac{1}{N_{pos}^c + N_{neg}^c}\Big(\sum_{k=1}^{N_{pos}^c} \rho(\boldsymbol{\pi}_{c_k}, \boldsymbol{r}_s) \cdot \log\psi(\boldsymbol{\pi}_{c_k}, \boldsymbol{r}_s)$$
$$+ \sum_{k=1}^{N_{neg}^c} \rho(\acute{\boldsymbol{\pi}}_{c_k}, \boldsymbol{r}_s) \cdot \log[1 - \psi(\acute{\boldsymbol{\pi}}_{c_k}, \boldsymbol{r}_s)]\Big) \quad (3)$$

where positive instance $\rho(\boldsymbol{x}_{u_m}, \boldsymbol{\pi}_c) = 1$ and negative instance $\rho(\acute{\boldsymbol{x}}_{u_m}, \boldsymbol{\pi}_c) = 0$. We minimize the joint mutual information-based loss $\mathcal{L}_{mu} = \lambda_c\mathcal{L}_{mu_c} + \lambda_g\mathcal{L}_{mu_g}$ to preserve the node-level user features, community-level and global graph-level dependency, , where $\lambda_c$ and $\lambda_g$ balance the information learned from $\mathcal{G}_c$ and $\mathcal{G}_s$. To this end, the social-aware user representations $\mathbf{X}^* \in \mathbb{R}^{M \times d}$ are generated while preserving the hierarchical social context.

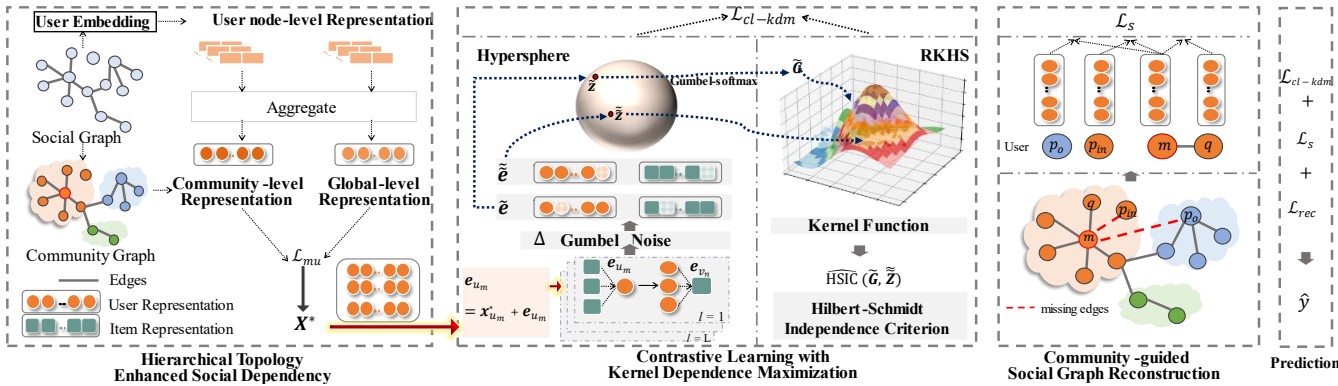

**Figure 2: The architecture of CL-KDM. (i) The hierarchical topology enhanced social dependency is built based on the node-community, and community-global mutual information maximization. Enhanced user representations X\* are injected into (ii); (ii) The graph contrastive learning with kernel dependence maximization is built based on the Hilbert-Schmidt Independence Criterion; (iii) The community-guided social graph reconstruction is to model partial order relationships among negative samples for social links.**

## 3.2 Graph Contrastive Learning with Kernel Dependence Maximization

Inspired by the success of SSL in vision tasks from a statistical dependence perspective [2], we propose a graph contrastive learning method based on the Hilbert Schmidt independence criterion to learn the kernel dependence maximization of augmented representations in high-dimensional space via kernel functions, improving the robustness of recommendation. We theoretically analyze that the difference of augmented representations in contrastive learning via subtle perturbations is limited by HSIC, and then we introduce the HSIC-based kernel dependence maximization as a regularizer to graph contrastive learning to improve recommendation.

*3.2.1* **Message Propagation** We adopt the GNNs as the backbone of our CL-KDM, and define the message propagation function as:

$$e_{u_m}^{(l+1)} = \sum_{n \in J_m} \frac{1}{\sqrt{|J_m||J_n|}} e_{v_n}^{(l)}$$

$$e_{v_n}^{(l+1)} = \sum_{m \in J_n} \frac{1}{\sqrt{|J_m||J_n|}} (e_{u_m}^{(l)} + x_{u_m}^*) \tag{4}$$

where $J_m$ represents the set of items that interact with user $u_m$, and $J_n$ denotes the set of users that interact with item $v_n$. $l$ denotes the index of graph neural network layers. $x_{u_m}^*$ is the social-aware user representation which is learned from our designed framework of hierarchical social dependency encoding (*Section 3.1*). Eq. (4) illustrates the concurrent incorporation of the node-level, community-level, and global-level social dependency into the process of message propagation.

*3.2.2* **Noise perturbations with Gumbel Distribution.** While the introduction of uniform perturbations to user/item nodes achieves representation-level augmentations and enhances the recommendation [3], uniformly distributed noise may cause the semantic inconsistency of augmented representations with the inputs. Inputting the Gumbel-distributed noise to representations can ensure that augmented representations do not compromise original semantics of a graph [1, 24]. Hence, we inject the Gumbel noise into representations to fortify the robustness of recommendation. Formally, given a user node $u_m$ and its representation $e_{u_m} \in \mathbb{R}^d$, we inject the Gumbel-distributed noises into $e_{u_m}$, and obtain the following augmented representations $\tilde{e}_{u_m}(\tilde{\tilde{e}}_{u_m})$:

$$\tilde{e}_{u_m} = e_{u_m} + \widetilde{\triangle}, \quad \tilde{\tilde{e}}_{u_m} = e_{u_m} + \widetilde{\widetilde{\triangle}}$$

$$s.t. \quad \|\triangle\|_2 + \epsilon, \quad \triangle + \triangle' \odot sign(e_{u_m}) \tag{5}$$

$$\triangle' = \log(-\log(\overline{\triangle})), \quad \overline{\triangle} \in \mathbb{R}^d \sim Uniform(0,1)$$

where $\widetilde{\triangle}$ and $\widetilde{\widetilde{\triangle}}$ denote the different Gumbel-distributed noise vectors $\triangle$ added to the same user node $u_m$, and $\epsilon$ controls the radius of a hypersphere [3]. We generate the augmented representations $\tilde{e}_{v_n}$ and $\tilde{\tilde{e}}_{v_n}$ of item $v_n$ using a similar way to that used for generating augmented user representations. At each layer $l$, different noise perturbations are imposed on the node embeddings [3]. For simplicity, we omit the superscript $l$.

*3.2.3* **Graph contrastive learning.** To mitigate the popularity bias problem in recommendation, we employ the InfoNCE loss [22] to learn each user representation by contrasting two different augmentations: $\mathcal{L}_{cl}^{user} = \sum_{m \in \mathcal{B}} -\log \frac{exp(\tilde{z}_{u_m}^{\mathrm{T}} \tilde{\tilde{z}}_{u_m}/\tau)}{\sum_{p \in \mathcal{B}/\{m\}} exp(\tilde{z}_{u_m}^{\mathrm{T}} \tilde{\tilde{z}}_{u_p}/\tau)}$, where $u_p$ is a user in a sampled batch $\mathcal{B}$, $\tilde{z}_{u_m}$ and $\tilde{\tilde{z}}_{u_m}$ are $L_2$ normalized $d$-dimensional representations learned from $\tilde{e}_{u_m}$ and $\tilde{\tilde{e}}_{u_m}$, and $\tau$ is the temperature [3]. Analogously, we obtain the contrastive loss of the item side $\mathcal{L}_{cl}^{item}$. We combine $\mathcal{L}_{cl}^{user}$ and $\mathcal{L}_{cl}^{item}$ to obtain graph contrastive learning loss $\mathcal{L}_{cl} = \mathcal{L}_{cl}^{user} + \mathcal{L}_{cl}^{item}$. $\mathcal{L}_{cl}$ achieves the alignment of same node representations and the divergence among different node representations. Meanwhile, $\mathcal{L}_{cl}$ enforces the uniformity of feature distribution on the unit hypersphere.

*3.2.4* **Hilbert-Schmidt Independence Criterion-based embedding learning:** Although $\mathcal{L}_{cl}$ learns representations by mapping the data onto a hypersphere via $L_2$ normalization, it cannot explicitly handle nonlinear dependence in high-dimensional space. Hence, we introduce HSIC [23] into graph contrastive learning to explicitly learn the kernel dependence between augmented representations in high-dimensional space. We use HSIC to measure the dependence between $\widetilde{Z}$ and $\widetilde{\widetilde{Z}}$ via performing a nonlinear transformation $\phi : \tilde{Z} \to \mathcal{F}$ and $\varphi : \tilde{\tilde{Z}} \to \mathcal{G}$ for each representation (Reproducing Kernel Hilbert Space (RKHS) with $\mathcal{F}$ and $\mathcal{G}$), and evaluating the norm of the cross-covariance between them:

$$HSIC(\widetilde{Z}, \widetilde{\widetilde{Z}}) = \left\| \mathbb{E}[\phi(\widetilde{Z})\varphi(\widetilde{\widetilde{Z}})^{\mathrm{T}}] - \mathbb{E}[\phi(\widetilde{Z})]\mathbb{E}[\varphi(\widetilde{\widetilde{Z}})]^{\mathrm{T}} \right\|_{HS}^2 \tag{6}$$

where $\|\cdot\|_{HS}^2$ is the Hilbert-Schmidt norm. HSIC identifies nonlinear dependence between $\widetilde{\mathbf{Z}}$ and $\widetilde{\widetilde{\mathbf{Z}}}$ with appropriate $\phi$ and $\varphi$ by measuring the scale of the correlation in these representations. Interestingly, through theoretical analysis [21, 50], we find that:

$$\text{HSIC}(\widetilde{\mathbf{Z}}, \widetilde{\widetilde{\mathbf{Z}}}) \geq \frac{M_{\mathcal{F}} M_{\mathcal{G}}}{M_{\widetilde{Z}} M_{\widetilde{\widetilde{Z}}}} \sup_{\theta} \text{Var}(h_{\theta}(\widetilde{\mathbf{Z}})) \tag{7}$$

where $h_{\theta}$ can be an encoder network, $M_{\mathcal{F}}$ and $M_{\mathcal{G}}$ denote the bounded of functions in $\mathcal{F}$, $\mathcal{G}$ (see Appendix). $M_{\widetilde{Z}}$ and $M_{\widetilde{\widetilde{Z}}}$ denote the bounded of functions in $\widetilde{Z}$ and $\widetilde{\widetilde{Z}}$. sup denotes the supremum. Eq (3) indicates that $\text{HSIC}(\widetilde{\mathbf{Z}}, \widetilde{\widetilde{\mathbf{Z}}})$ suppresses the variability of perturbations. Since $\widetilde{\mathbf{Z}}$ approximately satisfies the normal distribution $\widetilde{\mathbf{Z}} \sim \mathcal{N}(0, \sigma^2 \mathbf{I})$ after normalization, we can derive (proof see Appendix) [21]:

$$\frac{\xi \sqrt{-\log o(1)} \, dM_{\widetilde{Z}}}{\sigma M_{\mathcal{F}} M_{\mathcal{G}}} \text{HSIC}(\widetilde{\mathbf{Z}}, \widetilde{\widetilde{\mathbf{Z}}}) + o(\xi)$$
$$\geq \mathbb{E}\big[\big|h_{\theta}(\widetilde{\mathbf{Z}} + \Delta) - h_{\theta}(\widetilde{\mathbf{Z}})\big|\big], \Delta \in \mathbb{R}^d \tag{8}$$

where $o(1)$ stands for an arbitrary function $w: \mathbb{R} \to \mathbb{R}$ s.t. $\lim_{\xi \to 0} w(\xi) = 0$ [21]. If $\text{HSIC}(\widetilde{\mathbf{Z}}, \widetilde{\widetilde{\mathbf{Z}}}) = o(\frac{\sigma M_{\mathcal{F}} M_{\mathcal{G}}}{\sqrt{-2 \log o(1)} dM_{\widetilde{Z}}})$, then $\lim_{\xi \to 0} \sup_{\Delta \in \mathbb{R}^d} \mathbb{E}\big[\big|h_{\theta}(\widetilde{\mathbf{Z}} + \Delta) - h_{\theta}(\widetilde{\mathbf{Z}})\big|\big]/\xi = 0$. Assuming that $\widetilde{\widetilde{\mathbf{Z}}} = \widetilde{\mathbf{Z}} + \Delta$, and therefore, $\lim_{\xi \to 0} \sup_{\Delta \in \mathbb{R}^d} \mathbb{E}[|h_{\theta}(\widetilde{\widetilde{\mathbf{Z}}}) - h_{\theta}(\widetilde{\mathbf{Z}})|]/\xi = 0$, i.e., the difference between augmented representations is almost constant under small input perturbations. Hence, $\text{HSIC}(\widetilde{\mathbf{Z}}, \widetilde{\widetilde{\mathbf{Z}}})$ limits the variance between the augmented representations, which causes model sensitivity to input perturbations and poorly generalization to new datasets. We therefore add the HSIC as regularization term into $\mathcal{L}_{cl}$ to improve robustness.

*3.2.5* **Estimator of HSIC:** We need to correctly and efficiently estimate $\text{HSIC}(\widetilde{\mathbf{Z}}, \widetilde{\widetilde{\mathbf{Z}}})$. To alleviate the overfitting, we perform a Gumbel-softmax to $\widetilde{\mathbf{Z}}$ based on the Gumbel distribution:

$$\mathbf{W}_{\widetilde{Z}} = \text{Gumbel\_softmax}(\widetilde{Z})$$

$$\widetilde{\mathbf{G}} = \widetilde{\mathbf{Z}} \odot \mathbf{W}_{\widetilde{Z}} \tag{9}$$

where $\widetilde{\mathbf{G}}$ is fed into HSIC. We turn to estimate $\text{HSIC}(\widetilde{\mathbf{G}}, \widetilde{\widetilde{\mathbf{Z}}})$. Inner products in RKHS are calculated by functions: $\mathcal{K}(\widetilde{g}_{u_m}, \widetilde{g}_{u_m}{}') = \langle \phi(\widetilde{g}_{u_m}), \phi(\widetilde{g}_{u_m}{}') \rangle_{\mathcal{F}}$, $\mathcal{I}(\widetilde{\widetilde{z}}_{u_m}, \widetilde{\widetilde{z}}_{u_m}{}') = \langle \varphi(\widetilde{\widetilde{z}}_{u_m}), \varphi(\widetilde{\widetilde{z}}_{u_m}{}') \rangle_{\mathcal{G}}$, where $\mathcal{K}$ and $\mathcal{I}$ are kernel functions. Let $(\widetilde{g}_{u_m}, \widetilde{g}_{u_m}{}')$, $(\widetilde{\widetilde{z}}_{u_m}, \widetilde{\widetilde{z}}_{u_m}{}')$ be independent copies of $(\widetilde{g}_{u_m}, \widetilde{\widetilde{z}}_{u_m})$, and this gives:

$$\text{HSIC}(\widetilde{\mathbf{G}}, \widetilde{\widetilde{\mathbf{Z}}}) = \mathbb{E}_{\widetilde{G}, \widetilde{G}', \widetilde{\widetilde{Z}}, \widetilde{\widetilde{Z}}'} [\mathcal{K}(\widetilde{G}, \widetilde{G}') \mathcal{I}(\widetilde{\widetilde{Z}}, \widetilde{\widetilde{Z}}')] -$$
$$2\mathbb{E}_{\widetilde{G}, \widetilde{\widetilde{Z}}}[\mathcal{K}(\widetilde{G}, \widetilde{G}') \mathcal{I}(\widetilde{\widetilde{Z}}, \widetilde{\widetilde{Z}}')] + \mathbb{E}_{\widetilde{G}, \widetilde{G}'}[\mathcal{K}(\widetilde{G}, \widetilde{G}') \mathcal{I}(\widetilde{\widetilde{Z}}, \widetilde{\widetilde{Z}}')] \tag{10}$$

Given $T$ i.i.d. samples $\{(\widetilde{g}_1, \widetilde{\widetilde{z}}_1{}'), \dots, (\widetilde{g}_T, \widetilde{\widetilde{z}}_T{}')\}$ drawn i.i.d. from the joint distribution of $(\widetilde{\mathbf{G}}, \widetilde{\widetilde{\mathbf{Z}}})$, we estimate HSIC by: $\widehat{\text{HSIC}}(\widetilde{\mathbf{G}}, \widetilde{\widetilde{\mathbf{Z}}}) = (T-1)^{-2}\text{Tr}((\mathbf{K}(\widetilde{\mathbf{G}}, \widetilde{\mathbf{G}}')\mathbf{H}\mathbf{L}(\widetilde{\widetilde{\mathbf{Z}}}, \widetilde{\widetilde{\mathbf{Z}}}')\mathbf{H})$. The kernel matrices are denoted as $\mathbf{K}_{m,q} = \mathcal{K}(\widetilde{g}_{u_m}, \widetilde{g}_{u_q})$ and $\mathbf{L}_{m,q} = \mathcal{I}(\widetilde{\widetilde{z}}_{u_m}, \widetilde{\widetilde{z}}_{u_q})$, and $\mathbf{H} = \mathbf{I} - \frac{1}{T}\mathbf{1}\mathbf{1}^{\text{T}}$ is the centering matrix. The kernel functions $\mathcal{K}$ and $\mathcal{I}$ endow CL-KDM with the ability to map the representations in low-dimensional space to high-dimensional space [4]. This process changes the dimensionality and captures the nonlinear properties of data in high-dimensional space.

*3.2.6* **Kernel Dependence Maximization.** Our method extends the framework employed by most of recent SSL methods [2, 3, 21]. For an interaction graph $\mathcal{G}_r$ with $(N + M)$ nodes, each node $u_m$ ($v_n$) is augmented with noise perturbations, and then forms different augmented representation $\widetilde{\boldsymbol{e}}_{u_m}$ ($\widetilde{\boldsymbol{e}}_{v_n}$) and $\widetilde{\widetilde{\boldsymbol{e}}}_{u_m}$ ($\widetilde{\widetilde{\boldsymbol{e}}}_{v_n}$). We associate each node with its different augmentations. For example, to match $\widetilde{\boldsymbol{e}}_{u_m}$ and $\widetilde{\widetilde{\boldsymbol{e}}}_{u_m}$, we maximize the kernel dependence between augmented representation $\widetilde{\boldsymbol{g}}_{u_m}$ and $\widetilde{\widetilde{\boldsymbol{z}}}_{u_m}$ of the same node $u_m$. To construct representations tailored for recommendation tasks, we penalize high-variance representations. These ideas coalesce in our HSIC-based kernel dependence maximization objective for graph contrastive learning. We define the graph contrastive learning loss with kernel dependence maximization $\mathcal{L}_{cl-kdm}$ as:

$$\mathcal{L}_{kdm} = -\widehat{\text{HSIC}}(\widetilde{\mathbf{G}}, \widetilde{\widetilde{\mathbf{Z}}}) + \eta \, \widehat{\text{HSIC}}(\widetilde{\mathbf{G}}, \widetilde{\mathbf{G}})^{-\frac{1}{2}} + \eta_1 \widehat{\text{HSIC}}(\widetilde{\widetilde{\mathbf{Z}}}, \widetilde{\widetilde{\mathbf{Z}}})^{-\frac{1}{2}}$$

$$\mathcal{L}_{cl-kdm} = \lambda_{cl}\mathcal{L}_{cl} + \lambda_{kdm}\mathcal{L}_{kdm} \tag{11}$$

where $\eta$ and $\eta_1$ control the importance of $\widehat{\text{HSIC}}(\widetilde{\mathbf{G}}, \widetilde{\mathbf{G}})^{-\frac{1}{2}}$ and $\widehat{\text{HSIC}}(\widetilde{\widetilde{\mathbf{Z}}}, \widetilde{\widetilde{\mathbf{Z}}})^{-\frac{1}{2}}$. The square root for $\widehat{\text{HSIC}}(\widetilde{\widetilde{\mathbf{Z}}}, \widetilde{\widetilde{\mathbf{Z}}})$, $\widehat{\text{HSIC}}(\widetilde{\mathbf{G}}, \widetilde{\mathbf{G}})$ restricts them on the same scale. $\lambda_{kdm}$ is the regularization weight of kernel dependence maximization. $\widehat{\text{HSIC}}(\widetilde{\mathbf{G}}, \widetilde{\widetilde{\mathbf{Z}}})$ estimates the degree of agreement between the augment representations $\widetilde{\mathbf{G}}$ and $\widetilde{\widetilde{\mathbf{Z}}}$, which naturally leads to representation uniformity [2]. By minimizing the loss $\mathcal{L}_{kdm}$, we can maximize the kernel dependence between $\widetilde{g}$ and $\widetilde{\widetilde{z}}$ to match different augmented representations of the same node. HSIC improves the ability of representations to discriminate in high-dimensional space by using kernel functions.

## 3.3 Community-guided Social Graph Reconstruction

Enhancing the embedding propagation module by social relationship reconstruction can inject social-aware signals into recommendation. Most social-aware research reconstructs social signals through BPR pairwise loss [5]. However, these reconstruction tasks cannot learn the partial order relationships among negative samples (pseudo-positive samples). Motivated by the characteristics of community structures that users in the same community tend to generate more social relations than users in different communities [48], we propose a community structure-guided social graph reconstruction network to learn this partial order relationship for enhanced recommendation. Specifically, we reconstruct the adjacent matrix $\mathbf{A}_s$ of $\mathcal{G}_s$:

$$s_{m,q}^{A_s} = \delta(\mathbf{E}_{u_m}, \mathbf{E}_{u_q})$$

$$\mathcal{L}_{s1} = -\frac{1}{\psi(\mathbf{A}_s)} \sum_{\substack{(m,q) \in \mathcal{S} \\ (m,p_{in}) \notin \mathcal{S}}} \log \sigma(s_{m,q}^{A_s} - s_{m,p_{in}}^{A_s})$$

$$\mathcal{L}_{s2} = -\frac{1}{\psi(\mathbf{A}_s)} \sum_{\substack{(m,p_{in}) \in \mathcal{S} \\ (m,p_o) \notin \mathcal{S}}} \log \sigma(s_{m,p_{in}}^{A_s} - s_{m,p_o}^{A_s}) \tag{12}$$

where $\delta(\cdot)$ can either be a MLP or an inner product, and $\psi(\mathbf{A}_s)$ indicates the number of non-zero elements in $\mathbf{A}_s$. $\mathcal{S}$ denotes the observed user-user social links. $p_{in}$ indicates that user $u_{p_{in}}$ and user $u_m$ are in the same community without social links. Notably, $p_o$ indicates that user $u_{p_o}$ and user $u_m$ are in different communities without links. We define our joint loss function for the user-user reconstruction as: $\mathcal{L}_s = \mu_1\mathcal{L}_{s1} + \mu_2\mathcal{L}_{s2}$, where $\mu_1$ and $\mu_2$ are the hyperparameters to adjust the weight proportion of $\mathcal{L}_{s1}$

and $\mathcal{L}_{s2}$. This process allows for the effective rebuilding of social relationships by incorporating community information and distinguishing between positive and negative links within the community and outside the community, resulting in high-quality user embeddings.

## 3.4 Model Training

We define the loss in our prediction layer as:

$$\hat{y}_{m,n} = \mathbf{E}_{u_m}{}^T \mathbf{E}_{v_n}$$
$$\mathcal{L}_{rec} = -\sum_{(m,n)\in\mathcal{R},(m,j)\notin\mathcal{R}} \log \sigma(\hat{y}_{m,n} - \hat{y}_{m,j}) \quad (13)$$

where $\hat{y}_{m,n}$ is the predictive score of the interaction between user $u_m$ and item $v_n$. $\mathcal{R}$ denotes the observed user-item interactions. After incorporating the community structure-guided social graph reconstruction loss stated in *Section 3.3* and graph contrastive learning loss with kernel dependence maximization in *Section 3.2*, we define our joint loss function as below:

$$\mathcal{L} = \mathcal{L}_{rec} + \mathcal{L}_{cl-kdm} + \omega_s \mathcal{L}_s + \omega_\Theta \|\Theta\|_F^2 \quad (14)$$

where $\omega_s$ are hyperparameters to $\mathcal{L}_s$. To prevent the overfitting issue, the regularization term is used where $\omega_\Theta$ and $\Theta$ denote the weight and model parameters, respectively.

## 3.5 Complexity analysis

Next, we analyze the complexity of the proposed CL-KDM model. Let $B$ denote the batch size, $\mathcal{T}$ denote the node number in a batch. i) In the graph contrastive learning with kernel dependence maximization, the computation cost of $\mathcal{L}_{cl}$ for contrasting positive/negative samples is $O(Bd + B\mathcal{T}d)$. The computation cost of $\mathcal{L}_{kdm}$ needs $O(B^2d^2)$. ii) In the hierarchical topology enhanced social dependency module, the complexity for calculation is $O(Bd)$. iii) In the community-guided graph reconstruction module, the loss $\mathcal{L}_s$ requires $O(d)$ operations for each user-user pair, and the complexity of $\mathcal{L}_s$ is $O(2Bd)$. iv) In the training phase, the complexity of $\mathcal{L}_{rec}$ is $O(2Bd)$. Specific experimental analysis of computational time is demonstrated in Section 4.5.

## 4 EVALUATION

In this section, extensive experiments are conducted for model evaluation to answer the following research questions:

• **RQ1:** How does CL-KDM perform compared to state-of-the-art recommendation methods? • **RQ2:** How does each module contribute to CL-KDM? • **RQ3:** How is the robustness of CL-KDM? • **RQ4:** How do the hyperparameters affect CL-KDM? • **RQ5:** How efficient is the model CL-KDM?

## 4.1 Experimental Settings

*4.1.1* **Datasets and Evaluation Protocols.** We conduct performance validation on three widely-used social recommendation datasets: Yelp[1], Douban[2], and Ciao[3]. Table 1 lists the statistics of these datasets. We randomly split each dataset into training/validation/test sets (80%/10%/10%). NDCG@K and Recall@K are used to evaluate the performance [13]. Additionally, HR@K, MRR@K [26], and Precision@K [10] are used for comparison and the results are given in the Appendix. The value of K is set at 5, 10, 20. We rank all items to evaluate the performance on the test set [25].

---

[1] https://github.com/lcwy220/Social-Recommendation
[2] https://github.com/librahu/HIN-Datasets-for-Recommendation-and-Network-Embedding
[3] http://www.cse.msu.edu/~tangjili/trust.html

**Table 1: Statistics of datasets**

| Dataset | Users | Items | Interactions | Social Relations | Interaction Density | Relations Density |
|---|---|---|---|---|---|---|
| Yelp | 21,461 | 102,433 | 894,435 | 497,206 | 0.0407% | 0.1080% |
| Douban | 13,367 | 12,677 | 1,068,278 | 8,170 | 0.6304% | 0.0046% |
| Ciao | 7,375 | 106,797 | 208,486 | 111,781 | 0.0026% | 0.2055% |

*4.1.2* **Comparison Models.** We compare CL-KDM with state-of-the-art methods from different research lines: 1) **MF-based methods**: BPR [5], NeuMF [6]; 2) **GNNs-based methods**: NGCF [7], DGCF [8], DiffNet [9]; 3) **SSL-enhanced recommendation methods**: MHCN [10], SEPT [11], SGL [12], NCL [13], SimGCL [3], AutoCF [14], DCCF [15]. More details about the comparison baseline models are given in the Appendix.

*4.1.3* **Hyperparameter Settings**: We implement our CL-KDM with Pytorch and utilize the Adam optimizer for model parameter inference, with a learning rate of 0.001 and embedding dimension of 64. For a fair comparison, we fine-tune all the hyperparameters of baselines with the grid search. $\lambda_{kdm}$ is tuned from {1e-5, 1e-4, ..., 0.01}. In accordance with the optimal kernel function selection [46][47], we choose several kernel functions as candidate kernels, which include the Gaussian kernel $\mathcal{K}(x,y) = exp(-\|x-y\|^2/2\sigma^2)$, Linear kernel, Laplacian kernel, Polynomial kernel, and Sigmoid kernel [47]. The bandwidth $\sigma$ in the Gaussian kernel is tuned between {0.001,0.01, ..., 1}. More details about the hyperparameter settings are in the Appendix.

## 4.2 Overall Performance Comparison (RQ1)

Table 2 encapsulates the comprehensive performance of the comparison methods across three social recommendation datasets. The summarized observations are as follows:

**Obs.1: Superiority over SSL approaches**. CL-KDM consistently attains superior performance over baselines, including strong self-supervised approaches in all datasets. By learning the nonlinear kernel dependence in high-dimensional space into the graph contrastive learning framework through our kernel dependence maximization, CL-KDM surpasses existing SSL methods. Current representation-level augmentations are limited by the strength of the designed perturbations, making it difficult to maintain semantic consistency and diversity of the augmented representations. Structure-level augmentations use subgraph structures and tend to destroy important information about graph structures. Overall, the robustness and generalization of recommendation to cope with different scenarios can be improved by using perturbations with Gumbel distribution and introducing HSIC into graph contrastive learning.

**Obs.2: Benefits of data augmentations.** In comparison baselines, we observe that SSL methods, such as SimGCL and NCL, consistently outperform pure GNN-based models like DGCF and NGCF in most cases. This confirms the advantages of self-supervised data augmentations within the realm of social recommendation. Such performance gap between GNN-based models and SSL-enhanced models could be attributed to: i) the long-tailed distribution of interaction data forces user embeddings to be constantly updated in the direction of popular items; ii) the highly clustered property of the representation distribution brought about by the message-passing mechanism induces representational degradation [3].

**Table 2: Experimental results on Yelp, Douban, and Ciao datasets. All improvements are significant with $p$-value < 0.01.**

| Datasets | Metrics | BPR | NeuMF | NGCF | DGCF | DiffNet | SEPT | MHCN | NCL | SGL | DCCF | AutoCF | SimGCL | CL-KDM |
|---|---|---|---|---|---|---|---|---|---|---|---|---|---|---|
| Yelp | Recall@5 | 0.0179 | 0.0143 | 0.0196 | 0.0223 | 0.0239 | 0.0259 | 0.0258 | 0.0294 | 0.0289 | 0.0265 | 0.0261 | 0.0306 | **0.0337** |
| | NDCG@5 | 0.0208 | 0.0168 | 0.0228 | 0.0254 | 0.0261 | 0.0287 | 0.0308 | 0.0361 | 0.0355 | 0.0304 | 0.0284 | 0.0380 | **0.0409** |
| | Recall@10 | 0.0292 | 0.0252 | 0.0338 | 0.0365 | 0.0373 | 0.0407 | 0.0429 | 0.0486 | 0.0472 | 0.0464 | 0.0432 | 0.0506 | **0.0546** |
| | NDCG@10 | 0.0239 | 0.0199 | 0.0267 | 0.0290 | 0.0294 | 0.0339 | 0.0351 | 0.0399 | 0.0396 | 0.0329 | 0.0323 | 0.0426 | **0.0453** |
| Douban | Recall@5 | 0.0663 | 0.0551 | 0.0727 | 0.0735 | 0.0749 | 0.0798 | 0.0834 | 0.0837 | 0.0849 | 0.0828 | 0.0819 | 0.0915 | **0.0960** |
| | NDCG@5 | 0.1369 | 0.1131 | 0.1420 | 0.1411 | 0.1417 | 0.1591 | 0.1613 | 0.1638 | 0.1672 | 0.1670 | 0.1645 | 0.1762 | **0.1821** |
| | Recall@10 | 0.1099 | 0.0921 | 0.1197 | 0.1205 | 0.1195 | 0.1245 | 0.1281 | 0.1189 | 0.1255 | 0.1211 | 0.1205 | 0.1409 | **0.1453** |
| | NDCG@10 | 0.1370 | 0.1134 | 0.1438 | 0.1441 | 0.1428 | 0.1568 | 0.1593 | 0.1595 | 0.1669 | 0.1539 | 0.1564 | 0.1745 | **0.1789** |
| Ciao | Recall@5 | 0.0234 | 0.0185 | 0.0262 | 0.0264 | 0.0265 | 0.0289 | 0.0295 | 0.0299 | 0.0309 | 0.0290 | 0.0294 | 0.0326 | **0.0355** |
| | NDCG@5 | 0.0283 | 0.0224 | 0.0295 | 0.0304 | 0.0306 | 0.0341 | 0.0348 | 0.0372 | 0.0370 | 0.0357 | 0.0359 | 0.0383 | **0.0421** |
| | Recall@10 | 0.0384 | 0.0288 | 0.0421 | 0.0414 | 0.0430 | 0.0460 | 0.0455 | 0.0485 | 0.0487 | 0.0464 | 0.0477 | 0.0491 | **0.0519** |
| | NDCG@10 | 0.0323 | 0.0246 | 0.0336 | 0.0339 | 0.0341 | 0.0375 | 0.0381 | 0.0405 | 0.0409 | 0.0394 | 0.0376 | 0.0413 | **0.0444** |

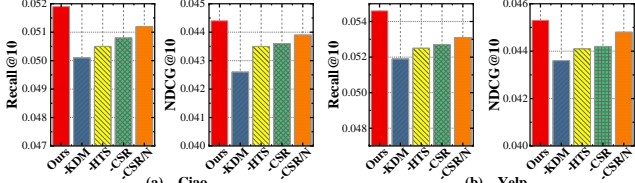

**Figure 3: Module ablation study on Ciao and Yelp.**

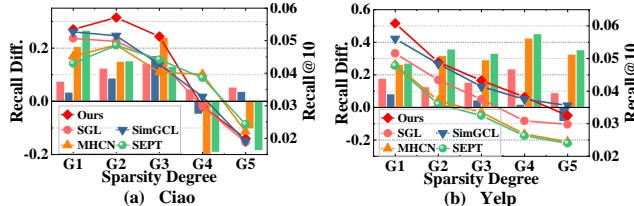

**Figure 4: Performance *w.r.t.* data sparsity. The bars represent the improvement ratio between CL-KDM and each baseline with the corresponding color, while the lines represent performance curves.**

## 4.3 Module Ablation Study (RQ2)

We conduct an ablation study to investigate the contribution of different sub-modules of CL-KDM in Figure 3.

**Effect of Kernel Dependence Maximization**. We investigate the benefit of kernel dependence maximization for the social recommendation task with the variant -KDM. In this variant, we remove the kernel dependence maximization $\mathcal{L}_{kdm}$. The results demonstrate an obvious improvement of our CL-KDM over the variant -KDM. This suggests that introducing HSIC into graph contrast learning and explicitly learn the kernel dependence between augmented representations can effectively regulate the representation uniformity in kernel Hilbert space to improve performance.

**Effect of Hierarchical Topology Enhanced Social Dependency Modeling**. We examine the advantage of the hierarchical topology enhanced social dependency modeling via the variant -HTS. In -HTS, we remove the hierarchical social-aware representations $\mathbf{X}^*$ for all users. The results in Figure 3 show that the variant -HTS reduces the performance of CL-KDM, demonstrating the advantages of learning hierarchical social dependency among users. Hence, the results indicate that the user representations are enhanced by injecting hierarchical social dependency into the embedding propagation process.

**Effect of Community-guided Social Graph Reconstruction**. In variant -CSR, the community-guided social graph reconstruction network $\mathcal{L}_s$ is removed. The results show that removing $\mathcal{L}_s$ causes performance degradation in recommendation and indicates the benefit of the community-guided social graph reconstruction module which enhances the user representations via injecting community-aware structural signals into the user-item graph. -CSR/N indicates that $\mathcal{L}_{s2}$ is removed. The results show that removing $\mathcal{L}_{s2}$ also affects performance, suggesting that social graph reconstruction can be better enhance recommendation by utilizing community structures to learn the partial order relationship of negative samples in $\mathcal{G}_s$.

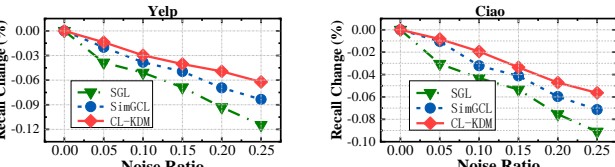

**Figure 5: Performance *w.r.t.* noise ratio.**

## 4.4 Model Robustness Test (RQ3)

**Robustness to Data Sparsity.** To validate the robustness of CL-KDM, we assess its performance on users with different sparsity levels in Figure 4. Users are categorized into five groups based on interaction frequency while keeping a consistent total interaction count within each group. We find that CL-KDM roughly surpasses these baselines. Meanwhile, on Ciao and Yelp, CL-KDM brings large performance gains mainly in scenarios with few interactions. This implies that CL-KDM can perform effective recommendation with sparse user-item interactions, benefited by the proposed graph contrastive learning with kernel dependence maximization.

**Robustness to Noisy Interactions.** We perform experiments to validate CL-KDM's robustness to noisy data. We taint the training set by adding different ratios of negative user-item interactions while leaving the testing set unaltered. As shown in Figure 5, the results exhibit that CL-KDM is robust under different ratios of noisy data, and it consistently outperforms the competitive baselines and mitigates the noise problem well. The superior performance of CL-KDM may stem from its injection of Gumbel noises into representations, ensuring the preservation of original semantics and effectively enhancing robustness.

## 4.5 Hyperparameter Investigation (RQ4)

In this section, we explore the impact of several important hyperparameters on the recommendation performance of CL-KDM. The evaluation results of Recall@10 and NDCG@10 are displayed in Figure 6, Figure 7, and Appendix. In Figure 7, the y-

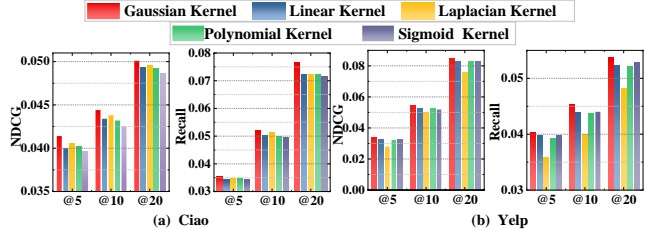

Figure 6: Influence of different kernel functions on performance.

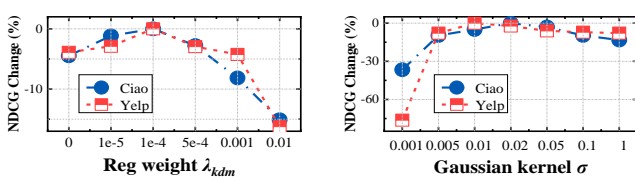

Figure 7: Hyperparameter study for CL-KDM in terms of NDCG@10 changes on Ciao and Yelp datasets.

axis denotes the relative performance degradation ratio compared to the optimal performance. We observe:

**Effect of kernel function selection.** The choice of kernel function is critical to the module of kernel dependence maximization, and different kernel functions can capture different types of nonlinear dependence in high-dimensional space. Here we test different Kernel functions and report their results in Figure 7. The results show that the best performance is achieved by using the Gaussian kernel function in the kernel dependence maximization module. This can be attributed to the smooth property of the Gaussian kernel function, and the smooth property makes CL-KDM more robust to noisy data.

**Effect of regularization weight $\lambda_{kdm}$ of kernel dependence maximization.** $\lambda_{kdm}$ determines the strength of kernel dependence maximization. As shown in Figure 7, an appropriate $\lambda_{kdm}$ improves the performance. However, further increasing $\lambda_{kdm}$ leads to a sharp decrease in performance. A larger $\lambda_{kdm}$ may cause excessive semantic deviation of the augmented representations from the original representations, thus damaging performance.

**Effect of Gaussian kernel bandwidth $\sigma$.** We study the influence of the bandwidth in the Gaussian kernel function on recommendation performance. The bandwidth aims to adjust the shape of the kernel function. By adjusting the bandwidth parameter, the complexity and generalization ability of the kernel function can be controlled to better accommodate different data distribution. The results in Figure 7 indicate that choosing a too-small bandwidth may cause overfitting, while a too-large bandwidth can lead to excessive smoothing, both negatively impacting performance.

### 4.6 Model Efficiency Study (RQ5)

**Model Convergence Analysis.** This section investigates the convergence of CL-KDM with different kernels and the results are depicted in Figure 8. The outcomes reveal an obviously faster convergence speed of our CL-KDM with the Gaussian kernel, underscoring its proficiency in training efficiency. Concurrently, it upholds an outstanding recommendation accuracy. Leveraging a Gaussian kernel expedites convergence in learning nonlinear dependence in high-dimensional space. Its expressive capacity

Table 3: Running time per epoch.

| Model | MHCN | AutoCF | DCCF | SGL | SimGCL | CL-KDM |
|-------|------|--------|------|-----|--------|--------|
| Yelp | 63.37s | 68.03s | 74.98s | 30.42s | 21.87s | 45.09s |
| Douban | 367.28s | 119.41s | 124.94s | 28.95s | 24.34s | 40.15s |
| Ciao | 10.06s | 27.98 s | 29.12s | 9.07s | 3.72s | 10.21s |

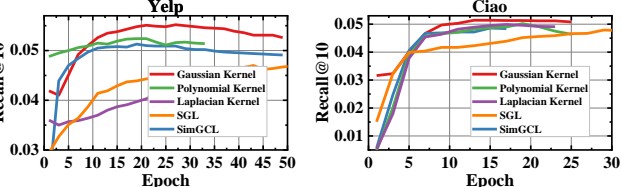

Figure 8: Convergence analysis.

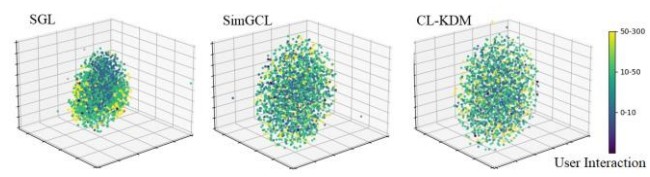

Figure 9: User embedding distribution using t-SNE on Ciao.

efficiently captures the nonlinear dependence and transforms data into linearly separable forms, thus accelerating convergence and enhancing training efficiency.

**Computational Cost Evaluation.** We further conduct a model efficiency analysis in terms of running time and present the evaluation results in Table 3. Although CL-KDM utilizes graph augmentations, the adjacency matrix of the graph needs to be generated only once before training, thus achieving competitive efficiency compared to other graph augmentation methods.

### 4.7 Embedding Visualization Analysis

In this section, we undertake an embedding visualization analysis using representations encoded by CL-KDM alongside competitive baselines. We map the learned representations (randomly sample 2,000 users for Ciao) to 3-D space using t-SNE [3]. As shown in Figure 9, the representations generated by CL-KDM exhibit uniformity and dispersion. In comparison to SGL, both SimGCL and CL-KDM achieve more uniformity. Furthermore, CL-KDM shows a significantly larger distance in embedding distribution, indicating that CL-KDM effectively preserves variability among representations while avoiding semantic deviation.

## 5 Conclusion

This work proposes a graph contrastive learning algorithm with kernel dependence maximization called CL-KDM to provide social recommendation with robustness and generalizability. CL-KDM can learn kernel dependence of augmented representations in high-dimensional space by introducing the Hilbert-Schmidt independence criterion into graph contrastive learning. Furthermore, hierarchical topology enhanced social dependency modeling as well as community-guided social graph reconstruction are integrated to enhance user embeddings. Extensive experiments are executed to validate the performance of CL-KDM. Results demonstrate the effectiveness of CL-KDM compared with existing recommendation methods.

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

# Appendix

## A.1 Theoretical Analysis

In this section, we give the theoretical analysis of Equation 7 and Equation 8 in the main text.

*A.1.1* **Derivation of equation (7):** Here, we formally provide the rationale for using HSIC to enhance robustness: we show theoretically that the regularization term $\text{HSIC}(\widetilde{\mathbf{Z}}, \widetilde{\widetilde{\mathbf{Z}}})$ makes the recommendation model less sensitive to input perturbations. Let $h_\theta = (g \circ f)$, where $g$ and $f$ denote the mapping function [21]. $\text{HSIC}(\widetilde{\mathbf{Z}}, \widetilde{\widetilde{\mathbf{Z}}})$ is associated with kernels $\mathcal{K}_{\widetilde{\mathbf{Z}}}, \mathcal{K}_{\widetilde{\widetilde{\mathbf{Z}}}}$. Let $\widetilde{\mathcal{Z}}, \widetilde{\widetilde{\mathcal{Z}}}$ be the supports of $\widetilde{\mathbf{Z}}, \widetilde{\widetilde{\mathbf{Z}}}$, respectively [50]. We assume that both $h_\theta$ and $g$ are continuous and bounded functions in $\widetilde{\mathcal{Z}}, \widetilde{\widetilde{\mathcal{Z}}}$, i.e., $h_\theta \in C(\widetilde{\mathcal{Z}}), g \in C(\widetilde{\widetilde{\mathcal{Z}}})$. Moreover, we assume that $h_\theta$ and $g$ are uniformly bounded, i.e., there are $0 < M_{\widetilde{\mathcal{Z}}}, M_{\widetilde{\widetilde{\mathcal{Z}}}} < \infty$ such that:

$$M_{\widetilde{\mathcal{Z}}} = \max_{h_\theta \in C(\widetilde{\mathcal{Z}})} \|h_\theta\|_\infty \text{ and } M_{\widetilde{\widetilde{\mathcal{Z}}}} = \max_{g \in C(\widetilde{\widetilde{\mathcal{Z}}})} \|g\|_\infty \quad (15)$$

We assume kernels $\mathcal{K}_{\widetilde{\mathbf{Z}}}$, $\mathcal{K}_{\widetilde{\widetilde{\mathbf{Z}}}}$ are universal with respect to $h_\theta$ and $g$, i.e., if $\mathcal{F}$ and $\mathcal{G}$ are the induced RKHS for kernels $\mathcal{K}_{\widetilde{\mathbf{Z}}}$, $\mathcal{K}_{\widetilde{\widetilde{\mathbf{Z}}}}$, respectively, then for uniformly bounded $h_\theta$, $g$ and any $\varepsilon > 0$ there are functions $h^{'} \in \mathcal{F}$ and $g^{'} \in \mathcal{G}$ such that $\|h_\theta - h^{'}\|_\infty \le \varepsilon$ and $\|g - g^{'}\|_\infty \le \varepsilon$. Moreover, functions in $\mathcal{F}$ and $\mathcal{G}$ are uniformly bounded, i.e., there exist $0 < M_{\mathcal{F}}, M_{\mathcal{F}} < \infty$ such that:

$$M_{\mathcal{F}} = \max_{f' \in \mathcal{F}} \|f'\|_\infty \text{ and } M_{\mathcal{G}} = \max_{g' \in \mathcal{G}} \|g'\|_\infty \quad (16)$$

Since $\mathcal{F}$, $\mathcal{G}$ are the two separable RKHS on $\widetilde{\mathcal{Z}}, \widetilde{\widetilde{\mathcal{Z}}}$ induced by $\mathcal{K}_{\widetilde{\mathbf{Z}}}$, $\mathcal{K}_{\widetilde{\widetilde{\mathbf{Z}}}}$. Then, the inequality holds: $\text{HSIC}(\widetilde{\mathbf{Z}}, \widetilde{\widetilde{\mathbf{Z}}}) \ge \sup_{s \in \mathcal{F}, t \in \mathcal{G}} \text{Cov}[s(\widetilde{\mathbf{Z}}), t(\widetilde{\widetilde{\mathbf{Z}}})]$. HSIC bounds the supremum of the covariance between functions in the RKHS. Let $\widetilde{\mathcal{F}} = \{h/M_{\mathcal{F}} : h \in \mathcal{F}\}$ and $\widetilde{\mathcal{G}} = \{g/M_{\mathcal{G}} : g \in \mathcal{G}\}$. In the original RKHS, we have:

$$\frac{M_{\widetilde{\mathcal{Z}}} M_{\widetilde{\widetilde{\mathcal{Z}}}}}{M_{\mathcal{F}} M_{\mathcal{G}}} \text{HSIC}(\widetilde{\mathbf{Z}}, \widetilde{\widetilde{\mathbf{Z}}}) = \sup_{s \in \widetilde{\mathcal{F}}, t \in \widetilde{\mathcal{G}}} \text{Cov}[s(\widetilde{\mathbf{Z}}), t(\widetilde{\widetilde{\mathbf{Z}}})] \quad (17)$$

According to [51, 50], we obtain:

$$\text{HSIC}(\widetilde{\mathbf{Z}}, \widetilde{\widetilde{\mathbf{Z}}}) \ge \frac{M_{\mathcal{F}} M_{\mathcal{G}}}{M_{\widetilde{\mathcal{Z}}} M_{\widetilde{\widetilde{\mathcal{Z}}}}} \sup_\theta \text{Var}[(h_\theta(\widetilde{\mathbf{Z}}))]$$

*A.1.2* **Derivation of equation (8):** Let $t_i : \mathbb{R}^d \to \mathbb{R}, i = 1, 2, \dots, d$ be the following truncation functions:

$$t_i(\widetilde{\mathbf{Z}}) = \begin{cases} -R, & \text{if } \widetilde{\mathbf{Z}}_i < R \\ \widetilde{\mathbf{Z}}_i, & \text{if } -R \le \widetilde{\mathbf{Z}}_i < R \\ R, & \text{if } \widetilde{\mathbf{Z}}_i > R \end{cases} \quad (18)$$

---

**Algorithm 1** CL-KDM

**Input**: user-item interaction graph $\mathcal{G}_r$, user-user social graph $\mathcal{G}_S$, user-community graph $\mathcal{G}_c$, maximum epoch numbers $P_1, P_2$, weight $\lambda_{kdm}$, $\omega_s$, bandwidth $\sigma$, learning rate $\eta$
**Output**: trained parameters in $\Theta$
**1 Initialize** all parameters in $\Theta$
**2 for** $e = 1$ to $P_1$ **do**
**3**      Calculate the community/global-level representation of $\mathcal{G}_c$, $\mathcal{G}_S$ (Eq 1)
**4**      Calculate the hierarchical mutual information-based loss $\mathcal{L}_{mu}$ (Eq 3)
**5**      **for** $\theta$ in the hierarchical social dependency modeling module **do**
**6**        $\theta = \theta - \eta \, \partial \mathcal{L}_{mu} / \partial \theta$
**7**      **end**
**8 end**
**9 for** $e = 1$ to $P_2$ **do**
**10**      inject the Gumbel-distributed noises $\widetilde{\Delta}$ into $\boldsymbol{e}_{u_m}, \boldsymbol{e}_{v_n}$
**11**      Calculate the Hilbert-Schmidt Independence Criterion $\widetilde{\text{HSIC}}(\widetilde{\mathcal{G}}, \widetilde{\widetilde{\mathbf{Z}}})$
**12**      Calculate the graph contrastive learning loss with kernel dependence maximization $\mathcal{L}_{cl-kdm}$ (Eq 11)
**13**      Draw a mini-batch of $(s_{m,q}^{A_s}, s_{m,p_{in}}^{A_s}, s_{m,p_n}^{A_s})$ for reconstruction
**14**      Calculate the reconstruction loss $\mathcal{L}_{s1}, \mathcal{L}_{s2}$ (Eq 12)
**15**      Calculate the predictive loss $\mathcal{L}_{rec}$ (Eq 13) and joint loss function $\mathcal{L}$ (Eq 14)
**16**      **for** $\theta$ in the Kernel Dependence Maximization module **do**
**17**        $\theta = \theta - \eta \, \partial \mathcal{L} / \partial \theta$
**18**      **end**
**19 end**
**20 return** all parameters $\Theta$

---

According to [21], we have

$$\frac{RM_{\widetilde{\widetilde{\mathcal{Z}}}}}{M_{\mathcal{F}} M_{\mathcal{G}}} \text{HSIC}(\widetilde{\mathbf{Z}}, \widetilde{\widetilde{\mathbf{Z}}}) \ge$$

$$\sup_{t \in C(\widetilde{\mathcal{Z}}) : \|t\|_\infty \le R, g \in C(\widetilde{\widetilde{\mathcal{Z}}}) : \|g\|_\infty \le M_{\widetilde{\widetilde{\mathbf{Z}}}}} \text{Cov}[t(\widetilde{\mathbf{Z}}), g(\widetilde{\widetilde{\mathbf{Z}}})] \ge \text{Cov}[\boldsymbol{t}_i(\widetilde{\mathbf{Z}}), \boldsymbol{h}_\theta(\widetilde{\mathbf{Z}})] \quad (19)$$

Let $\widetilde{\mathbf{Z}} \sim (0, \sigma^2 \mathbf{I})$ [49], we have:

$$\text{Cov}[\widetilde{\mathbf{Z}}_i, h_\theta(\widetilde{\mathbf{Z}})] - \text{Cov}[t_i(\widetilde{\mathbf{Z}}), h_\theta(\widetilde{\mathbf{Z}})] \le \frac{2M_{\widetilde{\mathcal{Z}}}\sigma}{\sqrt{2\pi}} \exp(-\frac{R^2}{2\sigma^2})$$

$$\text{Cov}[\widetilde{\mathbf{Z}}_i, h_\theta(\widetilde{\mathbf{Z}})] = \sigma^2 \mathbb{E}[\frac{\partial h_\theta(\widetilde{\mathbf{Z}})}{\partial \widetilde{\mathbf{z}}_i}] \quad (20)$$

Combining (19) and (20), we have:

$$\frac{RM_{\widetilde{\mathcal{Z}}}}{M_{\mathcal{F}} M_{\mathcal{G}}} \text{HSIC}(\widetilde{\mathbf{Z}}, \widetilde{\widetilde{\mathbf{Z}}}) + \frac{2M_{\widetilde{\mathcal{Z}}}\sigma}{\sqrt{2\pi}} \exp(-\frac{R^2}{2\sigma^2}) \ge \sigma^2 \mathbb{E}[\frac{\partial h_\theta(\widetilde{\mathbf{Z}})}{\partial \widetilde{\mathbf{z}}_i}] =$$

$$\sigma^2 \mathbb{E}[\sum_{i=1}^d |\frac{\partial h_\theta(\widetilde{\mathbf{Z}})}{\partial \widetilde{\mathbf{z}}_i}|] \quad (21)$$

By Taylor's theorem: we obtain:

$$\mathbb{E}[|h_\theta(\widetilde{\mathbf{Z}} + \Delta) - h_\theta(\widetilde{\mathbf{Z}})|] \le \mathbb{E}[|\Delta^{\mathrm{T}} \nabla_{\widetilde{\mathbf{Z}}} h_\theta(\widetilde{\mathbf{Z}})|] + o(\xi)$$

$$\le \xi \mathbb{E}[\sum_{i=1}^d |\frac{\partial h_\theta(\widetilde{\mathbf{Z}})}{\partial \widetilde{\mathbf{z}}_i}|] + o(\xi) \quad (22)$$

Let $R = \sigma\sqrt{-2\log o(1)}$. Then, we get $\frac{2\xi d}{\sqrt{2\pi}\sigma} \exp(-\frac{R^2}{2\sigma^2}) = o(\xi)$. In conclusion, we have:

$$\frac{\xi\sqrt{-2\log o(1)}dM_{\widetilde{\mathcal{Z}}}}{\sigma M_{\mathcal{F}} M_{\mathcal{G}}} \text{HSIC}(\widetilde{\mathbf{Z}}, \widetilde{\widetilde{\mathbf{Z}}}) + o(\xi)$$

$$\ge \mathbb{E}[|h_\theta(\widetilde{\mathbf{Z}} + \Delta) - h_\theta(\widetilde{\mathbf{Z}} + \Delta)|], \Delta \in \mathbb{R}^d$$

**Table 4: Experimental results on Yelp, Douban, and Ciao datasets in terms of HR, MRR, Precision.**

| Datasets | Metrics | BPR | NeuMF | NGCF | DGCF | DiffNet | SEPT | MHCN | NCL | SGL | DCCF | AutoCF | SIM GCL | CL-KDM |
|---|---|---|---|---|---|---|---|---|---|---|---|---|---|---|
| Yelp | HR@5 | 0.0666 | 0.0556 | 0.0746 | 0.0849 | 0.0846 | 0.1011 | 0.0994 | 0.0924 | 0.1131 | 0.1037 | 0.0997 | 0.1233 | **0.1280** |
| | MRR@5 | 0.0363 | 0.0303 | 0.0400 | 0.0440 | 0.0445 | 0.0543 | 0.0531 | 0.0534 | 0.0584 | 0.0556 | 0.0529 | 0.0675 | **0.0723** |
| | Precision@5 | 0.0142 | 0.0116 | 0.0163 | 0.0182 | 0.0197 | 0.0221 | 0.0227 | 0.0254 | 0.0243 | 0.0234 | 0.0219 | 0.0276 | **0.0292** |
| | HR@10 | 0.1086 | 0.0943 | 0.1234 | 0.1337 | 0.1399 | 0.1550 | 0.1571 | 0.1671 | 0.1647 | 0.1521 | 0.1522 | 0.1884 | **0.1931** |
| | MRR@10 | 0.0417 | 0.0354 | 0.0464 | 0.0504 | 0.0497 | 0.0614 | 0.0607 | 0.0608 | 0.0664 | 0.0609 | 0.0603 | 0.0755 | **0.0799** |
| | Precision@10 | 0.0122 | 0.0103 | 0.0143 | 0.0154 | 0.0155 | 0.0185 | 0.0195 | 0.0135 | 0.0201 | 0.0171 | 0.0199 | 0.0232 | **0.0245** |
| | HR@20 | 0.1638 | 0.1478 | 0.1887 | 0.2001 | 0.2095 | 0.2323 | 0.2326 | 0.2396 | 0.2405 | 0.2373 | 0.2314 | 0.2701 | **0.2747** |
| | Recall@20 | 0.0470 | 0.0418 | 0.0546 | 0.0572 | 0.0597 | 0.0697 | 0.0679 | 0.0711 | 0.0709 | 0.0674 | 0.0671 | 0.0799 | **0.0853** |
| | MRR@20 | 0.0455 | 0.0391 | 0.0509 | 0.0550 | 0.0534 | 0.0666 | 0.0659 | 0.0624 | 0.0614 | 0.0681 | 0.0685 | 0.0806 | **0.0856** |
| | NDCG@20 | 0.0287 | 0.0248 | 0.0328 | 0.0350 | 0.0365 | 0.0432 | 0.0424 | 0.0453 | 0.0449 | 0.0416 | 0.0406 | 0.0510 | **0.0537** |
| | Precision@20 | 0.0100 | 0.0086 | 0.0118 | 0.0127 | 0.0131 | 0.0162 | 0.0159 | 0.0169 | 0.0164 | 0.0160 | 0.0158 | 0.0185 | **0.0194** |
| Douban | HR@5 | 0.3774 | 0.3365 | 0.3864 | 0.3901 | 0.3953 | 0.4135 | 0.4191 | 0.4201 | 0.3982 | 0.4058 | 0.4101 | 0.4448 | **0.4546** |
| | MRR@5 | 0.2334 | 0.1963 | 0.2377 | 0.2391 | 0.2416 | 0.2662 | 0.2693 | 0.2566 | 0.2525 | 0.2607 | 0.2614 | 0.2896 | **0.2975** |
| | Precision@5 | 0.1162 | 0.0974 | 0.1190 | 0.1188 | 0.1181 | 0.1313 | 0.1319 | 0.1271 | 0.1266 | 0.1254 | 0.1247 | 0.1445 | **0.1495** |
| | HR@10 | 0.4968 | 0.4475 | 0.5105 | 0.5131 | 0.5130 | 0.5258 | 0.5309 | 0.5100 | 0.5275 | 0.5004 | 0.5010 | 0.5569 | **0.5689** |
| | MRR@10 | 0.2494 | 0.2113 | 0.2624 | 0.2631 | 0.2572 | 0.2812 | 0.2841 | 0.2775 | 0.2704 | 0.2710 | 0.2745 | 0.3044 | **0.3132** |
| | Precision@10 | 0.0986 | 0.0858 | 0.1015 | 0.1004 | 0.0999 | 0.1085 | 0.1093 | 0.1060 | 0.1058 | 0.1049 | 0.1504 | 0.1193 | **0.1224** |
| | HR@20 | 0.6068 | 0.5664 | 0.6198 | 0.6198 | 0.6240 | 0.6365 | 0.6357 | 0.6356 | 0.6314 | 0.6301 | 0.6355 | 0.6582 | **0.6692** |
| | Recall@20 | 0.1743 | 0.1502 | 0.1840 | 0.1834 | 0.1829 | 0.1923 | 0.1926 | 0.1969 | 0.1907 | 0.1911 | 0.1934 | 0.2115 | **0.2186** |
| | MRR@20 | 0.2571 | 0.2195 | 0.2699 | 0.2620 | 0.2651 | 0.2890 | 0.2915 | 0.2761 | 0.2753 | 0.2711 | 0.2716 | 0.3113 | **0.3195** |
| | NDCG@20 | 0.1467 | 0.1222 | 0.1535 | 0.1527 | 0.1525 | 0.1655 | 0.1668 | 0.1691 | 0.1587 | 0.1641 | 0.1609 | 0.1833 | **0.1874** |
| | Precision@20 | 0.0808 | 0.0694 | 0.0825 | 0.0823 | 0.0812 | 0.0873 | 0.0867 | 0.0866 | 0.0858 | 0.0849 | 0.0855 | 0.0950 | **0.0973** |
| Ciao | HR@5 | 0.0837 | 0.0665 | 0.0908 | 0.0919 | 0.0911 | 0.1017 | 0.1042 | 0.1117 | 0.1049 | 0.1009 | 0.1011 | 0.1074 | **0.1150** |
| | MRR@5 | 0.0503 | 0.0397 | 0.0513 | 0.0534 | 0.0531 | 0.0600 | 0.0618 | 0.0647 | 0.0653 | 0.0609 | 0.0613 | 0.0661 | **0.0713** |
| | Precision@5 | 0.0204 | 0.0158 | 0.0217 | 0.0219 | 0.0221 | 0.0241 | 0.0245 | 0.0253 | 0.0250 | 0.0250 | 0.0250 | 0.0252 | **0.0276** |
| | HR@10 | 0.1186 | 0.0929 | 0.1346 | 0.1314 | 0.1351 | 0.1463 | 0.1455 | 0.1456 | 0.1473 | 0.1432 | 0.1409 | 0.1474 | **0.1571** |
| | MRR@10 | 0.0557 | 0.0431 | 0.0571 | 0.0587 | 0.0591 | 0.0656 | 0.0674 | 0.0704 | 0.0709 | 0.0678 | 0.0669 | 0.0713 | **0.0766** |
| | Precision@10 | 0.0156 | 0.0122 | 0.0174 | 0.0170 | 0.0175 | 0.0185 | 0.0189 | 0.0195 | 0.0191 | 0.0189 | 0.0181 | 0.0193 | **0.0206** |
| | HR@20 | 0.1558 | 0.1304 | 0.1785 | 0.1760 | 0.1791 | 0.1977 | 0.2010 | 0.2027 | 0.1987 | 0.1933 | 0.1902 | 0.2018 | **0.2114** |
| | Recall@20 | 0.0533 | 0.0450 | 0.0600 | 0.0586 | 0.0607 | 0.0669 | 0.0691 | 0.0689 | 0.0691 | 0.0675 | 0.0661 | 0.0703 | **0.0765** |
| | MRR@20 | 0.0582 | 0.0456 | 0.0601 | 0.0618 | 0.0620 | 0.0691 | 0.0712 | 0.0636 | 0.0743 | 0.0693 | 0.0690 | 0.0751 | **0.0802** |
| | NDCG@20 | 0.0359 | 0.0289 | 0.0387 | 0.0387 | 0.0391 | 0.0434 | 0.0448 | 0.0465 | 0.0464 | 0.0454 | 0.0452 | 0.0473 | **0.0501** |
| | Precision@20 | 0.0110 | 0.0090 | 0.0125 | 0.0122 | 0.0126 | 0.0135 | 0.0138 | 0.0136 | 0.0140 | 0.0132 | 0.0135 | 0.0140 | **0.0149** |

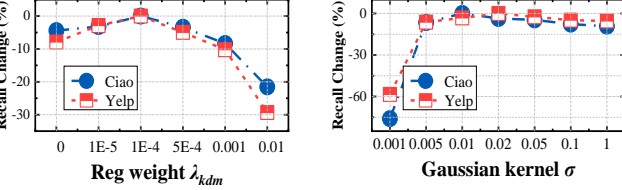

**Figure 10: Hyperparameter study for CL-KDM in terms of Recall@10 changes on Ciao and Yelp datasets.**

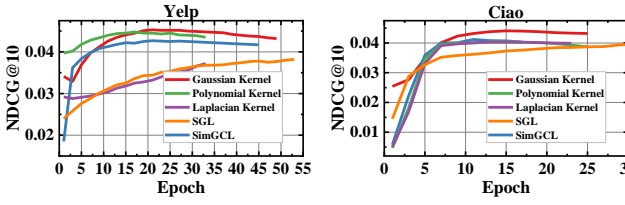

**Figure 11: Convergence analysis *w.r.t* epochs for training in terms of NDCG@10.**

## A.2 Supplementary Experiment Results

*A.2.1* **Comparison Model.** We compare CL-KDM with state-of-the-art methods from different research lines: **1) MF-based methods • BPR** [5]: A classical model that learns the latent representations with the matrix factorization (MF) framework. • **NeuMF** [6]: A representative neural collaborative filtering (CF) method that replaces the inner-product with a non-linear feature projection. **2) GNNs-based methods: • NGCF** [7]: It adopts the user-item graph to incorporate high-order connectivity for recommendation • **DGCF** [8]: It disentangles latent factors behind user-item interactions in a graph neural network architecture. • **DiffNet** [9]: A social recommendation model that mimics the social influence between different friends using graph attention networks. **3) SSL-enhanced recommendation methods: • MHCN** [10]: This model uses hypergraph convolution methods to capture different impacts of social motifs on preference learning among friends. • **SEPT** [11]: A social recommendation model which mines multiple positive samples with semi-supervised learning on the perturbed graph. • **SGL** [12]: It performs edge/node discarding to augment the graph data. • **NCL** [13]: A model that enhances graph contrastive learning through augmented structural and semantic neighbors. • **SimGCL** [3]: A simple CL method that adds uniform noises to the embedding space to create contrastive views. • **AutoCF** [14]: An automated

collaborative filtering that performs data augmentations for recommendation. • **DCCF** [15]: A framework that adaptively realizes disentanglement with self-supervised augmentations.

*A.2.2* **Hyperparameter Settings**: We set the temperature $\tau$ = 0.2. The batch size is selected from {512, 1024, 2048}. The number of propagation layers over the graph neural network is tuned from {1, 2, 3}. For a fair comparison, we refer to the best hyperparameter settings reported in the original papers of the baselines and then fine-tune all the hyperparameters of the baselines with the grid search. In our work, community information is constructed based on mining users' social links using the overlapping community detecting algorithm [48].

We list the experiment results on three datasets in terms of metrics HR@K, MRR@K, Precision@K at K=5, 10, 20 and NDCG@20, Recall@20 in Table 4. We display the experiment results of the hyperparameter study and convergence analysis on Yelp and Ciao datasets in Figure 10 and Figure 11. These results all show a similar trend to the results in the main text.