# OpenReview forum: "Graph Contrastive Learning with Kernel Dependence Maximization for Social Recommendation"
_ACM.org/TheWebConf/2024/Conference — TheWebConf24 Oral_

### Official Review · Reviewer_6zrb · 2023-10-31

**Novelty:** 6
**Technical Quality:** 6

**Review:**

Strengths:
- Novel contributions: The paper introduces a novel hierarchical topology enhanced social relation modeling method, as well as a HSIC-based kernel dependence maximization method. These contributions offer innovative approaches to address social relation modeling in the context of the proposed research.
- Theoretical analysis: The paper provides a theoretical analysis that demonstrates the limitations of representation-level augmentations by showing that they may be restricted to constants. This analysis supports the need for developing structure-level augmentations, highlighting the importance and significance of the proposed methodology.
- Comprehensive empirical study: The paper includes a comprehensive empirical study that covers various aspects such as overall performance comparison, module ablation study, robustness study, hyperparameter study, efficiency study, and analysis of the representation distribution. This thorough examination ensures a comprehensive understanding of the proposed method's effectiveness, efficiency, and efficacy in addressing the research problem.

Weaknesses:
- Confusing motivation for technical components: The authors attempt to highlight the motivation behind the two major technical components, namely the Hilbert-Schmidt independence criterion (HSIC) and the hierarchical social information modeling. However, the stated motivations seem to be somewhat confusing. For instance, in line 105, the first module is associated with the limitations of L2 normalization in InfoNCE, while in line 161, it is associated with the limitations of representation-level augmentations. It would be helpful if the authors could provide further clarification and coherence in the motivation explanations.
- Lack of clarity in community node and graph construction: The paper does not clearly explain the definition and construction of community nodes and the community graph. It would be beneficial for the authors to provide a detailed explanation of how the proposed method constructs social communities, including the criteria and techniques used for identifying and grouping nodes into communities. This clarification would enhance the readers' understanding of the proposed approach's social network modeling aspect.

**Questions:**

Please clarify the research motivation and the definition for the community nodes.

**Reviewer Confidence:**

3: The reviewer is confident but not certain that the evaluation is correct

**Scope:**

4: The work is relevant to the Web and to the track, and is of broad interest to the community

---

### Official Review · Reviewer_iPa7 · 2023-11-20

**Novelty:** 5
**Technical Quality:** 6

**Review:**

The document presents a graph contrastive learning method called CL-KDM, which aims to provide robustness and generalization ability for social recommendation. The method enhances user embeddings by introducing community information reconstruction networks and social dependency encoding. To address the challenges of existing methods, CL-KDM disentangles mutual information from the user-level to the community-level and from the community-level to the global-level, preserving the underlying structure. Furthermore, the method utilizes community-guided social graph reconstruction to further enhance user embeddings. Experimental results show that CL-KDM outperforms existing methods in terms of recommendation performance, generalization ability, and robustness.

**Questions:**

1.The method does not provide enough detail on how to integrate community structure into the model. The authors should provide more information on the model construction process to help readers better understand the method.
2.Although the experimental results demonstrate the superiority of CL-KDM in recommendation performance, a comparison with other methods in Figure 4 shows weakness of CL-KDM with higher sparsity. The authors should include more comparative experiment or give an analysis.
3.The article mentions the performance of CL-KDM in the presence of noisy data. To comprehensively evaluate the robustness of the model, the authors should add adversarial training or other relevant techniques to verify the stability of the model in resisting interference.
4.To improve the readability of the article, the authors should optimize the text expression, making it clearer and more accessible. Especially when describing model advantages and experimental results, the authors should use more intuitive ways to present the conclusions.
5.The article does not discuss the scalability and applicability of the model. The authors should analyze the performance of CL-KDM on different scales and types of datasets, as well as its potential in different application scenarios.

**Reviewer Confidence:**

3: The reviewer is confident but not certain that the evaluation is correct

**Scope:**

3: The work is somewhat relevant to the Web and to the track, and is of narrow interest to a sub-community

---

### Official Review · Reviewer_gQ4N · 2023-11-20

**Novelty:** 5
**Technical Quality:** 5

**Review:**

This work proposes a graph contrastive learning algorithm with kernel dependence maximization called CL-KDM to provide social recommendation with robustness and generalizability. CL-KDM can learn kernel dependence of augmented representations in high-dimensional space by introducing the Hilbert-Schmidt independence criterion into graph contrastive learning. Furthermore, hierarchical topology enhanced social dependency modeling as well as community-guided social graph reconstruction are integrated to enhance user embeddings.

**Questions:**

(1)The author should clarify the connection between "weak differentiation of representations in high-dimensional space," "information loss," and "nonlinear dependence" in the context of contrastive learning.
(2)Why is it claimed that contrastive learning cannot handle nonlinear dependence? Could the authors elaborate on this?
(3)How does HSIC address the issue of nonlinear dependence more effectively than previous methods?
(4)Why was NGCF chosen as a baseline for the experiments instead of a similar but more effective method like LightGCN?
(5)Can the author provide an explanation for the unexpected result in Table 2 where NGCF@5 outperforms NGCF@10 on the Douban dataset?

**Reviewer Confidence:**

3: The reviewer is confident but not certain that the evaluation is correct

**Scope:**

4: The work is relevant to the Web and to the track, and is of broad interest to the community

---

### Official Review · Reviewer_adUH · 2023-11-24

**Novelty:** 4
**Technical Quality:** 5

**Review:**

This paper studies the application of graph contrastive learning (CL) in social recommendation, highlighting issues in existing GCL methods such as L2 normalization problems and loss of structural information. The proposed CL-KDM integrates HSIC for kernel dependence maximization to tackle non-linear challenges and employs community-guided social graph construction to enhance user embeddings. Experiments are done on three datasets to demonstrate the effectiveness of CL-KDM.

Pros: The proposed CL-KDM outperforms existing contrastive learning methods significantly, showcasing its effectiveness. The use of HSIC space for kernel dependence maximization is a novel and intriguing aspect of the research.

Cons: The writing requires improvements. It lacks clear examples to elucidate complex concepts and tends to use overly complicated expressions. Additionally, providing further motivations behind each component of CL-KDM would better showcase their necessity and relevance.

**Questions:**

1. The rationale behind using HSIC is clear. However, the need for hierarchical, community-level structures is less so. Can you elaborate on the definition and significance of 'community' in this context? How are users grouped into communities? Could you provide examples to illustrate the impact of community-level structures on recommendations?

2. How does community-level message passing in your approach differ from existing hypergraph-based social recommendations?
It would be beneficial to discuss hypergraph-based social recommendation methods and compare them with CL-KDM to highlight its advantages.

3. Given the apparent similarity in function between the hierarchical topology-enhanced social dependency component and the community-guided social graph reconstruction component, can you provide a detailed explanation of how these two components differ in their approach and contribution to the model? What unique aspects does each component bring to CL-KDM, and how do these components interact or complement each other in improving recommendations?

4. As for the Experiments:
	a. It is noted that there are no baseline comparisons involving hypergraph-based recommendations, such as S2-MHCN [1]. Including such baselines could strengthen your evaluation.
	b. In Figure 9, the embeddings do not visibly demonstrate community characteristics in the t-SNE visualization. Can you explain this observation?

[1] Yu J, Yin H, Li J, et al. Self-supervised multi-channel hypergraph convolutional network for social recommendation[C]//Proceedings of the web conference 2021. 2021: 413-424.

**Reviewer Confidence:**

4: The reviewer is certain that the evaluation is correct and very familiar with the relevant literature

**Scope:**

4: The work is relevant to the Web and to the track, and is of broad interest to the community

---

### Official Review · Reviewer_YTJC · 2023-11-25

**Novelty:** 5
**Technical Quality:** 5

**Review:**

This paper proposed a graph contrastive learning model with kernal dependence maximization CL-KDM for social recommendation. The nonlinear kernel dependence of representations in Hilbert space is learnt by introducing a HSIC criterion. A hierarchical mutual information maximization module is designed for augmenting the user representations.  Experiments have been done over three real datasets to evaluate the performance of CL-KDM in terms of the recommendation quality.
Strength:
1. The paper is well motivated, and the research problem is interesting and practical.
2. CL-KDM is proposed and evaluated to be effective. The proposed technique is sound and solid.
3. 3 public available datasets are used in the evaluation.
Weakness:
1. In Related Work, the review on Social Recommendation is not complete. Many social recommendation approaches have been missed. The following papers are some examples for your discussion:
-- Real-time context-aware social media recommendation, VLDBJ 2019
-- Online Social Media Recommendation Over Streams, ICDE 2019
-- I-CARS: An Interactive Context-Aware Recommender System. ICDM 2019
-- Multi-Graph Heterogeneous Interaction Fusion for Social Recommendation. TOIS 2022
2. The presentation needs further improvement. There are some typos or grammatical errors. For example,
--In page 2, “Despite the encouraging results achieved by CL, however, they are not sufficient to …” please remove “however”
3. In Section 3, please put definitions and problem statement into separate paragraphs. Donot embed them into a single paragraph.
4. In A.2.2, Please discuss the reasons behand the performance change trend with parameter value change.
5. Please report the recommendation time cost as well.

**Questions:**

1. How is the performance comparison with non graph Contrastive Learning-based social recommendation?
2. How is the time cost of recommendation and model training?

**Reviewer Confidence:**

4: The reviewer is certain that the evaluation is correct and very familiar with the relevant literature

**Scope:**

4: The work is relevant to the Web and to the track, and is of broad interest to the community

---

### Decision · Program_Chairs · 2024-01-22

**Decision:**

Accept (Oral)

**Comment:**

This paper introduces a Graph Contrastive Learning method for social recommendation, where a kernelized approach is used in order to improve the quality of otherwise noisy embeddings learned through the contrastive process. The proposed method is theoretically grounded and appears to work well in practice.

 Overall, the reviews have been quite positive and supportive, and most concerns raised therein have been discussed and addressed to a satisfactory extent.